# Efficient Audiovisual Speech Processing via MUTUD: Multimodal Training and Unimodal Deployment

**Joanna Hong**                                                                joannahong@ieee.org
*Meta**

**Sanjeel Parekh**                                                              sanjeel@meta.com
*Meta*

**Honglie Chen**                                                                hongliechen@meta.com
*Meta*

**Jacob Donley**                                                                jdonley@meta.com
*Meta*

**Ke Tan**                                                                      tanke1116@meta.com
*Meta*

**Buye Xu**                                                                     xub@meta.com
*Meta*

**Anurag Kumar**                                                                anuragkr@ieee.org
*Meta**

**Reviewed on OpenReview:** *https: // openreview. net/ forum? id= 5bshBY8RDf*

## Abstract

Building reliable speech systems often requires combining multiple modalities, like audio and visual cues. While such multimodal solutions frequently lead to improvements in performance and may even be critical in certain cases, they come with several constraints such as increased sensory requirements, computational cost, and modality synchronization, to mention a few. These challenges constrain the direct uses of these multimodal solutions in real-world applications. In this work, we develop approaches where the learning happens with all available modalities but the deployment or inference is done with just one or reduced modalities. To do so, we propose a Multimodal Training and Unimodal Deployment (MUTUD) framework which includes a Temporally Aligned Modality feature Estimation (TAME) module that can estimate information from missing modality using modalities present during inference. This innovative approach facilitates the integration of information across different modalities, enhancing the overall inference process by leveraging the strengths of each modality to compensate for the absence of certain modalities during inference. We apply MUTUD to various audiovisual speech tasks and show that it can reduce the performance gap between the multimodal and corresponding unimodal models to a considerable extent. MUTUD can achieve this while reducing the model size and compute compared to multimodal models, in some cases by almost 80%.

---

*Work done while at Meta, now at Google DeepMind

# 1  Introduction

Unimodal (audio-only) approaches to well-known speech problems such as speech enhancement, speaker separation, and automatic speech recognition, have made rapid progress using deep learning. At the same time, multimodal approaches to these tasks are also increasingly gaining significance (Mira et al., 2023; Xu et al., 2020; Ma et al., 2021b; Hong et al., 2022; 2023). While the additional modality may come in different forms such as text, contact microphones, IMUs, etc., visual modality is the most widely used in these speech tasks. This bears similarity to humans as we also innately rely on visuals to perceive sounds and speech (Schwartz et al., 2004). In fact, people with hearing impairments have also been shown to rely on visuals for better perception of speech (Burnham et al., 2013). Given the significance of multimodal perception of speech by humans, it is natural that multimodal learning has shown impressive gains over unimodal systems for various speech tasks. The role of visuals in speech understanding becomes much more critical in acoustically difficult scenarios such as noisy environments or situations where the speech signals on their own are not reliable for the task at hand (Weninger et al., 2015; Tan & Wang, 2019; Wang et al., 2020; Braun et al., 2021).

While multimodal systems can extract supplementary and complementary information from different modalities (Baltrušaitis et al., 2018; Lu, 2023), leading to performance improvements, certain challenges with multimodal models can restrain their uses in real-world systems. These include but are not limited to *(1)* Multimodal models are often computationally much more expensive compared to their unimodal counterparts and the performance gain might not justify the substantial increase in computational cost. This is especially relevant for real-time and on-device applications (e.g., speech enhancement). In fact, in several cases, this can prohibit the deployment of multimodal systems. *(2)* Multimodal data comes at a significantly higher cost. Acquisition of multimodal data requires complex sensory devices working together seamlessly. Alignment, synchronization, and annotation efforts in multimodal data are far more challenging than audio-only data. More importantly, such aligned and synchronized multimodal data is required even during inference, necessitating the availability of all sensory devices and the processing power to align and synchronize the captured signals. This can make multimodal systems impractical in several real-world applications. *(3)* Lastly, it might not be feasible to use multiple modalities for a speech task due to practical constraints such as privacy or difficulties in getting signals for all modalities. For example, while multimodal ASR could improve audio-only ASR in noisy conditions, getting the visual signals during real-world uses might not be possible.

The above discussion highlights benefits of multimodal learning over unimodal learning, yet there are certain constraints which can make unimodal models preferable over multimodal despite lower performance. Motivated by this, the primary question we ask is *how do we learn from multimodal data while enabling unimodal uses of the model?* In this framework, we still want to learn from the rich information available in multimodal data but unimodal inference removes the constraints around uses of the multimodal system. Note that, unlike works on robustness to missing modality we develop a fundamental approach for *MUltimodal Training and Unimodal Deployment (MUTUD, pronounced "muted")*. In modality robustness, the model behavior remains the same during training and inference, and hence the challenges of multimodal systems outlined before are not rectified. MUTUD is driven by architectural and training novelties, which addresses those challenges. MUTUD framework is built using a novel *Temporally Aligned Modality feature Estimation (TAME)* module. The TAME module is designed to estimate deep representations of modalities which are *absent* during inference using the representations of modalities *present* during inference. TAME achieves this by having codebooks for each modality and linking cross-modal pairs of codebooks in a way that enables modality feature recall using the codebooks and the features of available modalities. Keeping in mind the importance of temporal information, TAME is designed to temporally align modalities sampled at different time resolutions.

We apply our framework for 3 well-known tasks in the speech processing domain and do multimodal (audiovisual (AV)) training and unimodal inference; speech enhancement, speech recognition, and active speaker detection. Speech enhancement in particular may have tight real-time and low-compute requirements for several applications. In all the tasks, we show that MUTUD achieves unimodal inference with a significantly better performance compared to the counterpart models trained on unimodal data. Moreover, compared to the full multimodal systems, our model has significantly lesser parameters and compute and yet gives competitive performance.

## 2 Related Works

**Audiovisual speech processing.** Analogous to humans, AV learning for speech-related tasks naturally results in methods that are more robust to noisy scenarios such as acoustic SNR degradation, poor lighting conditions, motion blur, etc. In this paper we focus on three AV speech problems namely, speech enhancement (Gabbay et al., 2017; Afouras et al., 2018a; Gao & Grauman, 2021; Mira et al., 2023; Yang et al., 2022; Owens & Efros, 2018; Hou et al., 2018), speech recognition (Huang & Kingsbury, 2013; Mroueh et al., 2015; Noda et al., 2015; Stewart et al., 2013; Ma et al., 2021b) and speaker detection (Garg et al., 2000; Cutler & Davis, 2000; Chakravarty et al., 2016; Roth et al., 2020). The reader is referred to excellent survey papers for a detailed overview of different methodologies (Michelsanti et al., 2021; Potamianos et al., 2017). As already highlighted, traditional AV approaches suffer from several constraints such as sensor requirements, computational cost, and modality synchronization which limit their applicability in real-world applications.

**Resource-constrained learning.** Considerable progress has been made in resource-constrained audio-only speech processing (Kim et al., 2020; Lee et al., 2021; Maayah et al., 2023), even though such multimodal methods are relatively smaller. Typical strategies include lightweight network design (Maayah et al., 2023), quantization and pruning (Tan et al., 2021) and knowledge distillation (Thakker et al., 2022). Gogate et al. (2020) build a robust language-dependent audiovisual model called CochleaNet for real-time speech enhancement through audiovisual mask estimation. LAVSE (Chuang et al., 2020) proposed a visual data compression technique for speech enhancement. Our focus in this work is very different. We intend to develop efficiency in multimodal learning by allowing resource-heavy modalities to be absent during prediction or when deployed.

**Learning with missing modality.** Multimodal learning for robustness to missing modality is a practical problem that has been explored in some works before. Each work differs in the modality considered to be missing, the phase (training or testing) in which this information is absent, and whether the loss of information is partial or complete (Hegde et al., 2021; Ma et al., 2021a; Chang et al., 2022; Woo et al., 2023; Ma et al., 2022; Lee et al., 2023). The methods are often tailor-made for the scenarios in consideration. For brevity, here we limit our discussion to AV speech-related tasks. Some studies rely on a memory architecture to retrieve missing modality via associated bridging mechanism (Kim et al., 2021b; Hong et al., 2021; Kim et al., 2021a). These related works serve as inspiration for MUTUD. Further, AV-HuBERT (Shi et al., 2022) and u-HuBERT (Hsu & Shi, 2022) presented a self-supervised pre-training framework that can leverage both multimodal and unimodal speech with a unified masked cluster prediction objective, achieving zero-shot modality generalization for multiple speech processing tasks. While these works have made significant progress on various speech processing problems, they are very different from ours – their focus is on self-supervised training of large models with massive amounts of unlabeled data. The learned models are then fine-tuned for tasks like ASR. These models are not designed for unimodal deployment with compute/memory efficiency in mind. Furthermore, it is difficult to adapt AV-HuBERT/u-HuBERT for tasks like speech enhancement, especially in causal settings.

Unlike these works, we are driven by the challenges of multimodal learning outlined before. We focus explicitly on multimodal learning for unimodal prediction and real-world deployment, which addresses those challenges. Our approach is fairly generic and can be applied to many common multimodal learning methods and tasks.

## 3 MUTUD: Multimodal Training and Unimodal Deployment

We describe our proposed method, which we call Multimodal Training and Unimodal Deployment (MUTUD). Our goal is to design a network that leverages multimodal sensory inputs during training, but only takes in a subset of them during inference. In section 3.1, we first describe MUTUD in its general setting, where an arbitrary number of modalities are considered, followed by a discussion targeted to the audiovisual speech domain. In section 3.2, we introduce our proposed TAME Module, which is the key component to enable unimodal predictions. Finally in section 3.3, we describe the training objectives. The left panel in Figure 1 shows the difference between MUTUD and conventional multimodal learning.

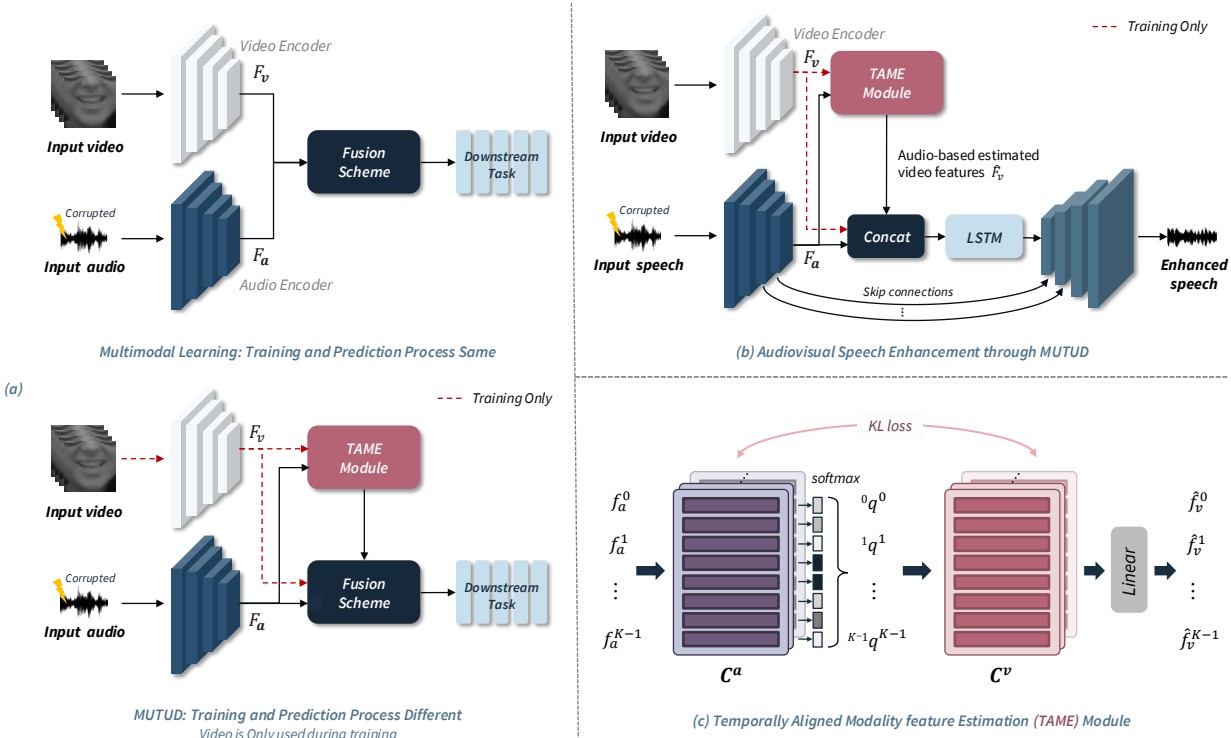

Figure 1: (a) The left panel shows a comparison between conventional audiovisual speech processing and MUTUD. TAME module enables audiovisual learning without doing video processing during prediction. (b) The upper half in the right panel illustrates MUTUD for an AVSE model. After training the video encoder is discarded. (c) The bottom half in the right panel shows the estimation of video representations using TAME. The illustration is for $t = 0$ in Eq 5.

## 3.1 MUTUD Overview

Let $\mathcal{D}$ be a dataset where each sample $X \in \mathcal{D}$ is characterized by $M$ different modalities $X = \{X_{m_i}\}$, $i = 1 : M$. $\mathcal{M} = \{m_1, m_2, \cdots, m_M\}$ is the set of modalities. Conventionally, multimodal learning operates with the assumption that the model always inputs all $M$ modalities, during training as well as for predictions. Let $h^{\mathcal{M}}(X = \{X_{m_i}, m_i \in \mathcal{M}\}; \phi)$ a deep neural network (DNN) based multimodal system (parameterized by $\phi$). In MUTUD, all $M$ modalities of $X$ are available during training, but only a subset, $\mathcal{M}_s \subset \mathcal{M}$, are available during real-world deployment or for inference.

To this end, we design MUTUD with two crucial characteristics in mind. Let $h(; \theta)$ be the MUTUD system. (1) Since $h(; \theta)$ processes only $|\mathcal{M}_s|$ modalities for prediction, we expect it to have fewer parameters and be computationally more efficient in real-world deployment. Ideally, we would like $h(; \theta)$ to have inference size and compute similar to $h^{\mathcal{M}_s}(X = \{X_{m_i}, m_i \in \mathcal{M}_s\}; \psi)$, a model counterpart of $h^{\mathcal{M}}(X; \phi)$ with $\mathcal{M} = \mathcal{M}_s$. (2) On the performance end, $h^{\mathcal{M}}(; \phi)$ should have superior performance compared to $h^{\mathcal{M}_s}(; \psi)$ due to utilization of more modalities in the learning process. We expect $h(; \theta)$ to have superior performance compared to $h^{\mathcal{M}_s}(X; \psi)$ and closer to that of $h^{\mathcal{M}}(X; \phi)$.

In a typical multimodal model, all $X_{m_i}$ are encoded by a network, these representations are then fused through various mechanisms (concatenation, attention, etc. Kalkhorani et al. (2023); Wei et al. (2020); Ma et al. (2021b); Lee et al. (2020); Praveen et al. (2023)). The fused representations are further processed by more neural layers to solve the task at hand. We operate in a similar setting. To achieve our goal, we develop an efficient and effective mechanism to *associate* and *relate* missing modalities, $(\mathcal{M} - \mathcal{M}_s)$, to those in $\mathcal{M}_s$, such that the representations of $X_{m_i} \in \mathcal{M} - \mathcal{M}_s$ can be recalled using those of $X_{m_i} \in \mathcal{M}_s$. We propose a Temporally Aligned Modality feature Estimation (TAME) module. TAME learns a pair of codebooks ($\boldsymbol{C}^{m_i}$,

$C^{m_j}$) for each pair of modality in $\{(m_i, m_j), \forall m_i \in \mathcal{M} - \mathcal{M}_s, \forall m_j \in \mathcal{M}_s\}$. The training objectives link these codebooks in a way that enables estimation of representations for $X_{m_i} \in \mathcal{M} - \mathcal{M}_s$ during inference.

**MUTUD for AudioVisual Speech Processing.** We focus on audiovisual speech tasks where MUTUD is designed to use only one of them during deployment. For a succinct and clear description of MUTUD and TAME, we explain it through the task of Audiovisual Speech Enhancement (AVSE) but the method similarly adapts to other tasks. The right panel in Figure 1 outlines the base AVSE model ($h^{\mathcal{M}}(;\phi)$). The speech and video encoders produce $F_a \in \mathbb{R}^{T_a \times D}$ and $F_v \in \mathbb{R}^{T_v \times D}$ representations, respectively. $T_a$ and $T_v$ represent time dimensions and depend on the frame rates of speech and audio. The frame rate of speech is $K$ times of video ($T_a = T_v * K$) and hence $F_v$ is upsampled by a factor of $K$ to match the size along a temporal direction before the concatenation step. The concatenated representations are then decoded by the decoder to produce the enhanced speech. The $h^{\mathcal{M}_s}(;\psi)$ model is the Audio-only model, where everything is the same except that there are no visual inputs, and the decoder decodes the encoded audio representations to output enhanced speech. Under MUTUD, our goal is to train with both visual and audio inputs but deploy an Audio-only model. Hence, we design and train TAME module to estimate video representations during prediction.

## 3.2 TAME Module

The core of the TAME consists of modality-specific codebooks (MSCs) for audio and video. These are used to associate and relate modalities through their respective representations during training. During inference, the audio representations are used to retrieve the video representations through these MSCs. The MSCs are designed to capture temporal alignment and synchronized relations between the audio and the video. Since the audio representations frame rate (in $F_a$) is higher by a factor of $K$, we design TAME keeping this temporal relation in consideration. That is the $t^{th}$ video frame feature in $F_v$, $f_v^t$, is associated with $K$ audio features ($f_a^{K \cdot t}, f_a^{K \cdot t+1}, \ldots, f_a^{K \cdot t+K-1}$) in $F_a$. Besides keeping the temporal alignment between audio and video representations intact, this temporal coupling between the audio and video is also necessary for learning to estimate video features using audio.

TAME formulates this through $K$ blocks of codebooks in each MSC, represented as $C^a \in \mathbb{R}^{K \times N \times D}$ and $C^v \in \mathbb{R}^{K \times N \times D}$ for the audio and video respectively, (see Figure 1). $N$ is the number of codes in each set of codebooks in $C^a$ and $C^v$.

All features in consideration ($f_v^t$ for video and $\boldsymbol{f}_a^t = \{f_a^{K \cdot t}, f_a^{K \cdot t+1}, \ldots, f_a^{K \cdot t+k}, \ldots, f_a^{K \cdot t+K-1}\}$ for audio) are first embedded through their respective MSC. This relationship between $f_v^t$ and $k^{th}$ codebook in $C^v$ is established through the vectors $^k v^t$. For improved readability, we represent the left superscripts $(k)$ $^k v^t$ vectors as $\dot{v}^t$

$$\dot{v}^t = \frac{\langle \dot{c}_n^v, f_v^t \rangle}{\|\dot{c}_n^v\|_2 \, \|f_v^t\|_2} \, , \tag{1}$$
$$\text{where } \dot{c}_n^v = C^v[k, n, :], \; n = \{0, 1, \ldots, N\}.$$

$\dot{v}^t$ is computed for all $K$ codebooks ($k \in \{0, K-1\}$) using Eq 1. Similarly, the audio features are related to its codebooks $C^a$ as,

$$\dot{a}^{K \cdot t+k} = \frac{\langle \dot{c}_n^a, f_a^{K \cdot t+k} \rangle}{\|\dot{c}_n^a\|_2 \, \|f_a^{K \cdot t+k}\|_2} \, , \tag{2}$$
$$\text{where } \dot{c}_n^a = C^a[k, n, :], \; n = \{0, 1, \ldots, N\}$$

The temporal steps $t$ are $\{0, 1, \ldots, T_v - 1\}$. Note that, for audio the $k^{th}$ codebook of $C^a$ is linked with $k^{th}$ audio feature in $\boldsymbol{f}_a^t$. Eq 1 and 2 embed the audio and video information into their respective MSCs. A softmax across the number of codes gives the probability distribution of the relationship between the codebooks and the corresponding modality representations,

$$\dot{p}^t = \frac{\exp\left(\tau \cdot \dot{v}_n^t\right)}{\sum_{j=1}^{N} \exp\left(\tau \cdot \dot{v}_j^t\right)} , \tag{3}$$

$$\dot{q}^{K \cdot t+k} = \frac{\exp\left(\tau \cdot \dot{a}_n^{K \cdot t+k}\right)}{\sum_{j=1}^{N} \exp\left(\tau \cdot \dot{a}_j^{K \cdot t+k}\right)} , \tag{4}$$

$$\text{where } n = \{0, 1, \ldots, N\}$$

$\tau$ is the temperature for the softmax function. These distributions are computed for each $k \in \{0, 1, \ldots, K-1\}$. The modality-specific information captured by $\dot{p}^t$ and $\dot{q}^t$ are used to relate and associate the two modalities as well as retrieve the video representations using the audio representations.

**Audio-to-Video Representations.** The bottom half in the right panel of Figure 1 shows the schematics for obtaining video representations using audio. The $k^{th}$ feature in $\boldsymbol{f}_a^t$ directly estimates "interleaved" representations for video using the $k^{th}$ codebook in $\boldsymbol{C}^v$,

$$\hat{f}_v^{K \cdot t+k} = \text{linear}\left(\sum_{n=1}^{N} \dot{q}_n^{K \cdot t+k} . \dot{c}_n^v; \ \theta_l\right) \tag{5}$$

where $\theta_l$ are the parameters of the linear layer, in practice, this linear layer includes batch-normalization (Ioffe & Szegedy, 2015). The $\hat{f}_v^{K \cdot t+k}$ (instead) are concatenated with $f_a^{K \cdot t+k}$ and then decoded by the decoder to produce enhanced speech. Note that, in the base AVSE model $T_a$ video features are simply repeated to upsample by a factor of $K$ and then concatenated to audio features. TAME helps estimate video information at a lower temporal resolution, which can be crucial for precise replacement of video representations.

Clearly, the video encoder is discarded during inference and as long as the size and compute of the TAME module is significantly smaller than the video encoder, the whole model is much more efficient compared to the full audiovisual model.

### 3.3 Training Objectives

**TAME Specific Losses.** To train the proposed TAME module, we propose three different training objectives. First, we need to ensure that the relationship between the video features and video codebook $\boldsymbol{C}^v$ is well-structured so that $\boldsymbol{C}^v$ gets embedded with video information. This is achieved through self-modality recall of $f_v^t$ for each codebook in $\boldsymbol{C}^v$, ${}^k\tilde{f}_v^t = \text{linear}(\sum_{n=1}^{N} \dot{p}_n^t . \dot{c}_n^v; \theta_l)$. A reconstruction loss then guides the learning

$$\mathcal{L}_{v \to v} = \sum_{t=0}^{T_v-1} \sum_{k=0}^{K-1} \| {}^k\tilde{f}_v^t - f_v^t \|_2^2 \tag{6}$$

Next, a reconstruction loss between the estimated video representations $\hat{f}_v^{K \cdot t+k}$ and $f_v^t$ enforces retrieval of video information through audio representations.

$$\mathcal{L}_{a \to v} = \sum_{t=0}^{T_v-1} \sum_{k=0}^{K-1} \| \hat{f}_v^{K \cdot t+k} - f_v^t \|_2^2 \tag{7}$$

Lastly, we establish a cross-modal association by linking the two MSCs through the distribution captured by $\dot{p}^t$ and $\dot{q}^{K*t+k}$. Let $P^k$ (captured by $\dot{p}^t$) and $Q^k$ (captured by $\dot{q}^{K*t+k}$) be the distributions over the codes for $k^{th}$ codebook in $\boldsymbol{C}^v$ and $\boldsymbol{C}^a$ respectively.

$$\mathcal{L}_{C_a \to C_v} = \sum_{k=1}^{K} D_{KL}(P^k || Q^k). \tag{8}$$

The loss function in Eq 8, the distribution of codes in each codebook of $C^a$ matches the corresponding ones in $C^v$. This is necessary as the codebooks in $C^v$ are probed using audio representations embedded in $\dot{q}^t$ to obtain video representations.

**Task-Specific Loss Functions.** The overall training of MUTUD includes task-specific loss functions which in this case are speech enhancement losses. In this work, the outputs of the enhancement models are complex spectrograms ($E$) of the enhanced speech. The time-domain waveform ($e$) from $E$ is obtained using the Inverse-Short Time Fourier Transform. The speech enhancement loss functions we use are

$$\mathcal{L}_{\text{task}} = \|E - C\|_1 - \text{SI-SDR}(e, c) \tag{9}$$

where $C$ is the complex STFT of target clean speech and $c$ is the time-domain target clean speech. The SI-SDR loss is defined as $\text{SI-SDR}(e, c) = 10 \log_{10} \frac{\|\alpha c\|^2}{\|\alpha c - e\|^2}$, where $\alpha = \frac{e^T c}{\|c\|^2}$. The enhancement losses in Eq 9 are computed using both $f_v^t$ and $\hat{f}_v^t$ as inputs to the decoder and the overall $L_{\text{task}}$ is the sum of these losses. This is necessary to warrant that the video encoder learns meaningful representations in the end-to-end training.

The total loss function is

$$\mathcal{L}_{\text{MUTUD}} = \mathcal{L}_{v \to v} + \mathcal{L}_{a \to v} + \mathcal{L}_{C_a \to C_v} + \lambda \mathcal{L}_{\text{task}} \tag{10}$$

where $\lambda$ is the weight given to the task loss.

A few points are worth noting here. The TAME which is enabling MUTUD seamlessly fits into the base AVSE framework and can be easily adopted for many common multimodal methods and tasks. In our experiments, we evaluate MUTUD for 3 multimodal tasks; AVSE, audiovisual speech recognition (AVSR), and audiovisual active speaker detection (AV-ASD).

## 4 Experimental Setup

We evaluate MUTUD under 3 multimodal tasks; AVSE, audiovisual speech recognition (AVSR), and ego-centric audiovisual active speaker detection (AV-ASD). AVSE is of key focus as this task is often desired to be deployed in real-time communication and on-device, which exacerbates the multimodal challenges outlined earlier in the paper.

### 4.1 Datasets

For AVSE and AVSR tasks, we utilize the LRS3-TED corpus (Afouras et al., 2018b), a large-scale audiovisual dataset for speech tasks. For the AV-ASD task, we use *EasyCom*, a challenging real-world egocentric dataset (Donley et al., 2021). Overall, this allows for a comprehensive evaluation of MUTUD under a wide variety of acoustic and visual noise conditions.

**LRS3-TED.** LRS3-TED corpus (Afouras et al., 2018b) is a large-scale dataset of TED and TEDx videos. LRS3-TED consists of audio-visual pairs and corresponding text transcriptions for 151,819 utterances, totaling 439 hours. Following the original splits, we use $\sim$131,000 utterances for training and $\sim$1,300 utterances for testing. For AVSE, the clean speech samples are taken from LRS3 and the noise samples are taken from Reddy et al. (2021) noise set. The videos are 25 fps with $224 \times 224$ resolution. During pre-processing, we center-crop at the mouth with a size of $88 \times 88$.

**EasyCom.** We employ EasyCom (Donley et al., 2021) for the AV-ASD task. This dataset contains $\sim 5$ hours of natural conversations recorded in a noisy restaurant-like environment. The ego-centric nature of the data makes it extremely challenging as the sensory devices (camera and microphone on wearable glasses) are always moving. The ego-motions make it difficult to learn from the video and the audio is corrupted by noise, making audiovisual active speaker detection (AV-ASD), challenging on this dataset. The dataset includes annotated voice activity, speech transcriptions, head bounding boxes, target of speech, and source identification labels. We use train-test splits from Hsu et al. (2022).

## 4.2 Implementation Details for AVSE

**Data processing.** For LRS3, we crop the lip regions, resize the cropped frames into 88×88, and transform them to grayscale following Kim et al. (2021c). The audio, sampled at 16kHz, is converted into a spectrogram using a window size of 20 ms and a hop length of 10 ms. We augment the video data by applying random spatial erasure and time masking for effective modeling of the visual context (Mira et al., 2022).

All models are trained using noisy-clean speech pairs where, speech samples from LRS3 are mixed with noise samples from the DNS Challenge (Reddy et al., 2021) noise set. The noisy mixture is obtained by randomly mixing up to 5 different noise samples. The SNR range for mixing is -15 dB to 10 dB. We report results under 2 test conditions, (a) 3 background noises (3-BN) are present in the noisy mixture, and (b) 5 background noises (5-BN) are present. Evaluations are done at five different SNRs (in dB): 5, 0, -5, -10, and -15.

**Architectural and Training details.** We use **3** different backbones of audio-only/audiovisual models for comprehensively evaluating our proposed MUTUD on the speech enhancement task. 2 of the backbone models are inspired from the U-Net architecture design of gated convolutional recurrent network (GCRN) (Tan & Wang, 2019) and the corresponding audiovisual model (Mira et al., 2023). The Audio-only enhancement model here is a U-Net style encoder-decoder architecture. The input to the audio model is complex spectrogram of the audio. The audio encoder is composed of stacking of 4 gated convolutional blocks; which consists of two 2D convolutional layers, where the outputs of each convolutional layer, one followed by Sigmoid activation, are multiplied. The decoder includes an LSTM layer. The Audiovisual model (Mira et al., 2023) is built on top of this Audio-only model by employing a 3D convolutional layer followed by a ResNet18 (He et al., 2016) as the video encoder. The video and audio encoder outputs are concatenated and forwarded through the decoder to produce complex spectrograms of the enhanced audio. The concatenated audio features and video features are taken into a 2-layer Grouped LSTM. The decoder consists of 5 deconvolutional layers with a skip connection like a U-net architecture. The encoder-decoder structure is designed in a symmetric way, where the number of kernels progressively increases in the encoder and decreases in the decoder. To aggregate the context along the frequency direction, a stride of 2 is adopted along the frequency, dimension in all convolutional and up-convolutional layers. For MUTUD, $K = 4$ and we set the number of codes, $N$ in the MSCs to 32 after conducting an ablation study for different $N$ (Sec. 5.5). We train using AdamW optimizer (Kingma & Ba, 2014) with a learning rate of $10^{-4}$. We adopt a cosine scheduler (Loshchilov & Hutter, 2016), adding a warmup for 20 epochs. Loss function hyperparameter $\lambda$ is set to 0.01.

We also use VisualVoice (Gao & Grauman, 2021) as another type of backbone for audiovisual speech enhancement and follow the original architecture details and implementations.

**Evaluation metrics.** We utilize three standard speech quality and intelligibility metrics for AVSE: Short Time Objective Intelligibility (STOI) (Taal et al., 2010), Scale-Invariant Signal-to-Distortion Ratio (SISDR) (Le Roux et al., 2019), Perceptual Evaluation of Speech Quality (PESQ) (Rix et al., 2001).

## 5 Results and Discussions

### 5.1 Effectiveness of MUTUD

Table 1 presents quantitative results for the AVSE task under 3-background noise test conditions. A few important details about the reported methods are in order. For a fair comparison, in addition to Audio-only, we also report the performance of Audio-only (*w. matched params*), that is, a model with the number of parameters matched with MUTUD. This is important to establish that the proposed TAME module is in fact providing crucial information not present in the audio modality and cannot be compensated for by simply adding more parameters to the Audio-only model. We show results for two versions of MUTUD representing two different training mechanisms: One where we train the entire model from scratch, denoted by *w.o. pretrained TAME*. Another is where we first pre-train the TAME module solely with clean audio and video frames and then fine-tune the entire model for the enhancement task. This is done to better guide the TAME module to store modality-specific information in the MSCs.

It is clear from Table 1 that our proposed framework MUTUD outperforms both the Audio-only and the Audio-only *with matched parameters* over all metrics and SNRs. This shows that the model has learned

Table 1: Speech Enhancement performance comparison of different models for 3-BN test condition.

| Method | STOI (%) | | | | | SISDR (dB) | | | | | PESQ | | | | |
|---|---|---|---|---|---|---|---|---|---|---|---|---|---|---|---|
| | 5 | 0 | -5 | -10 | -15 | 5 | 0 | -5 | -10 | -15 | 5 | 0 | -5 | -10 | -15 |
| **Noisy Audio** | 82.6 | 72.4 | 60.5 | 48.8 | 38.9 | 5.00 | 0.01 | -5.02 | -10.03 | -15.07 | 1.24 | 1.12 | 1.07 | 1.07 | 1.07 |
| **Audio-only** | 92.7 | 88.1 | 80.1 | 67.5 | 51.5 | 13.64 | 10.55 | 7.08 | 2.88 | -2.82 | 2.18 | 1.80 | 1.48 | 1.27 | 1.14 |
| **Audio-only** *w. similar # params* | 93.0 | 88.3 | 80.4 | 68.0 | 52.6 | 13.75 | 10.58 | 7.05 | 2.86 | -2.62 | 2.30 | 1.87 | 1.53 | 1.30 | 1.16 |
| **MUTUD** *w.o. pretrained TAME* | 93.4 | 89.1 | 81.6 | 69.5 | 53.6 | 14.07 | 10.99 | 7.54 | 3.32 | -2.24 | 2.37 | 1.93 | 1.58 | 1.33 | 1.17 |
| **MUTUD** | **93.5** | 89.2 | 81.8 | 69.8 | 54.0 | **14.11** | **11.02** | 7.56 | 3.38 | -2.19 | **2.36** | 1.92 | 1.57 | 1.32 | 1.17 |
| **Audiovisual** | **93.5** | **89.6** | **83.3** | **74.0** | **62.7** | 13.92 | 10.90 | **7.61** | **3.81** | **-0.86** | 2.35 | **1.94** | **1.60** | **1.38** | **1.20** |

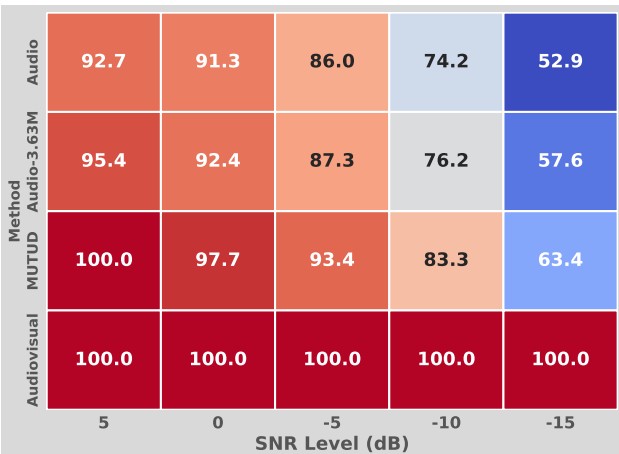

Figure 2: MUTUD bridges the gap between Audiovisual and Audio-only models. Performance (in %) of gain in STOI through different methods relative to the gain *Audiovisual* brings in average intelligibility (STOI) of noisy speech samples. MUTUD is able to get most of performance gains of the *Audiovisual* model across different SNRs. For example, at -5dB SNR *Audio-only* gives gives 86.0% of the gain of the Audioivisual model whereas MUTUD is at 93.4% of Audiovisual model.

with visual information available during training and MUTUD is able to estimate video encodings and use them for better enhancement. It is worth mentioning that for extremely low SNRs of -10dB and -15dB, where multimodal models heavily rely on visual information for speech enhancement, MUTUD continues to consistently perform better than the Audio-only model. Figure 2 shows each model's performance relative to the Audiovisual model, the gain Audiovisual model brings in speech intelligibility (STOI) at different SNRs. As one can see at each SNR, the MUTUD model is able to recover most of the Audiovisual performance, without directly using visual signal during inference. This further highlights the TAME module's ability to estimate relevant visual information at prediction time. While we do not expect the MUTUD model to outperform or fully match the performance of the Audiovisual model, it does an excellent job of reducing the gap between the unimodal and multimodal models. Except for extremely low SNR (-15dB), MUTUD is fairly competitive with the Audiovisual model on all 3 metrics. This further argues for our multimodal training and unimodal deployment strategy. We also observe that the pre-trained TAME module is slightly superior to the one simply trained from scratch.

We conduct additional experiments to further verify the effectiveness of MUTUD. We tackle a more challenging condition with a 5-background noise test. Shown in Table 2, we observe similar trends for the 5-background noise test conditions showing that MUTUD can be successfully employed in such extreme noise conditions.

**Spectrogram Visualization**: We visualize the spectrogram of each model to illustrate the improvements over the baseline in Figure 3. The advantage of MUTUD over Audio-only is most visible at SNR $-10/-15$ dB:

Table 2: Speech Enhancement performance comparison of different models for 5-BN test condition.

| Method | STOI (%) | | | | | SISDR (dB) | | | | | PESQ | | | | |
|---|---|---|---|---|---|---|---|---|---|---|---|---|---|---|---|
| | 5 | 0 | -5 | -10 | -15 | 5 | 0 | -5 | -10 | -15 | 5 | 0 | -5 | -10 | -15 |
| **Noisy Audio** | 81.7 | 70.8 | 58.2 | 46.0 | 36.3 | 5.00 | 0.00 | -5.00 | -10.00 | -15.02 | 1.21 | 1.10 | 1.06 | 1.06 | 1.08 |
| **Audio-only** | 92.2 | 87.0 | 78.3 | 64.4 | 47.0 | 13.28 | 10.08 | 6.47 | 1.99 | -4.17 | 2.16 | 1.74 | 1.43 | 1.23 | 1.11 |
| **MUTUD** *w.o. pretrained TAME* | 92.7 | 87.7 | 79.3 | 65.5 | 48.1 | 13.45 | 10.32 | 6.72 | 2.27 | -3.88 | 2.24 | 1.8 | 1.47 | 1.25 | 1.12 |
| **MUTUD** | 92.8 | 88.0 | 79.6 | 65.6 | 48.0 | **13.60** | **10.43** | 6.85 | 2.33 | -3.86 | 2.23 | 1.80 | 1.47 | 1.25 | 1.12 |
| **Audiovisual** | **92.9** | **88.6** | **81.6** | **71.3** | **58.9** | 13.43 | 10.37 | **6.95** | **2.90** | **-2.23** | **2.25** | **1.85** | **1.53** | **1.30** | **1.16** |

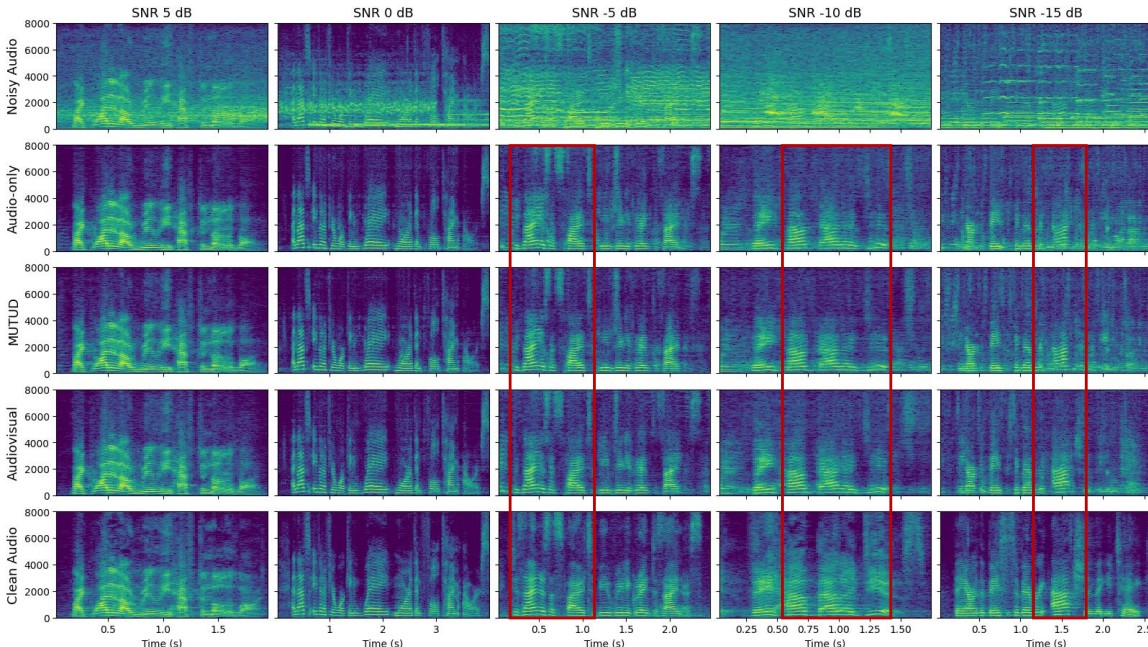

Figure 3: Spectrograms for noisy audio, *Audio-only*, MUTUD, *Audiovisual*, and clean audio (rows) at SNR = 5, 0, −5, −10, −15 dB (columns).

MUTUD preserves thinner, more continuous harmonics and darker inter-harmonic valleys (lower noise floor), and it shows a cleaner F0 trajectory with reduced low-frequency clutter. At −5 dB, MUTUD further reduces high-frequency speckle and keeps consonant transitions (short vertical stripes) sharper than Audio-only. Across all SNRs, MUTUD remains consistently closer to Audiovisual than Audio-only.

**Different Backbone Models**: We analyze the robustness of the TAME module with different architectural backbones. The first one is a smaller Audio-only architectural backbone inspired again from the GCRN (Tan & Wang, 2019) Audio-only model. To do so, we reduce the output dimension of the last two layers of the the audio encoder from 128 to 64. The visual part of the corresponding Audiovisual model remains same as before. This results in Audio-only and Audiovisual models with 0.815M and 12.195M parameters, respectively. Results in Table 3 verify the robustness of TAME module which showcases quantitative trends similar to those observed before, MUTUD is able to bridge the gap between the Audio-only model and the Audiovisual model. Finally, we adopt a different baseline, VisualVoice (Gao & Grauman, 2021), in order to demonstrate our method's flexibility with the underlying network architectures for the AVSE task. We successfully verify that, shown in Table 4, our proposed model can be adapted to a different baseline model achieving superior performance to the Audio-only model.

Table 3: Speech Enhancement performance comparison for Backbone 2. A smaller backbone, that is, the Audio-only and the corresponding audiovisual models used here are smaller and weaker.

| Method | STOI (%) | | | | | SISDR (dB) | | | | | PESQ | | | | |
|---|---|---|---|---|---|---|---|---|---|---|---|---|---|---|---|
| | 5 | 0 | -5 | -10 | -15 | 5 | 0 | -5 | -10 | -15 | 5 | 0 | -5 | -10 | -15 |
| **Noisy Audio** | 82.6 | 72.4 | 60.5 | 48.8 | 38.9 | 5.00 | 0.00 | -5.0 | -10.03 | -15.06 | 1.24 | 1.12 | 1.07 | 1.07 | 1.07 |
| **Audio-only** | 91.9 | 86.6 | 77.7 | 64.5 | 49.3 | 12.98 | 9.68 | 5.99 | 1.58 | -4.14 | 2.12 | 1.73 | 1.44 | 1.24 | 1.13 |
| **MUTUD** *w.o. pretrained TAME* | 92.3 | 87.3 | 78.9 | 66.0 | 50.3 | 13.31 | 10.07 | 6.45 | 2.08 | -3.65 | 2.20 | 1.79 | 1.49 | 1.27 | 1.14 |
| **MUTUD** | 92.3 | 87.3 | 78.9 | 66.0 | 50.5 | **13.29** | 10.04 | 6.43 | 2.06 | -3.66 | **2.22** | **1.81** | 1.49 | 1.27 | 1.14 |
| **Audiovisual** | **92.6** | **88.2** | **81.1** | **71.1** | **59.9** | 13.23 | **10.10** | **6.65** | **2.72** | **-2.01** | 2.15 | 1.80 | **1.51** | **1.31** | **1.18** |

Table 4: Speech Enhancement performance comparison for Backbone 3. VisualVoice as backbone for Audio-only, audiovisual, and MUTUD.

| Method | STOI (%) | | | | | SISDR (dB) | | | | | PESQ | | | | |
|---|---|---|---|---|---|---|---|---|---|---|---|---|---|---|---|
| | 5 | 0 | -5 | -10 | -15 | 5 | 0 | -5 | -10 | -15 | 5 | 0 | -5 | -10 | -15 |
| **Noisy Audio** | 82.6 | 72.4 | 60.5 | 48.8 | 38.9 | 5.00 | 0.00 | -5.00 | -10.03 | -15.06 | 1.24 | 1.12 | 1.07 | 1.07 | 1.07 |
| **Audio-only** | 91.8 | 85.9 | 76.5 | 64.0 | 48.8 | 11.67 | 8.62 | 5.17 | 1.08 | -4.90 | 2.05 | 1.60 | 1.32 | 1.17 | 1.09 |
| **MUTUD** | 93.5 | 89.0 | 82.0 | 71.3 | 56.5 | 12.72 | 9.68 | 6.53 | 2.98 | -2.04 | 2.40 | 1.94 | 1.58 | 1.33 | 1.18 |
| **Audiovisual** | **94.0** | **90.1** | **84.1** | **75.4** | **64.3** | **12.92** | **9.94** | **6.90** | **3.54** | **-0.86** | **2.51** | **2.03** | **1.65** | **1.39** | **1.21** |

## 5.2 Generalization to Audio-only Datasets

Our MUTUD models as well as the Audio-only and Audiovisual models are trained and evaluated on the LRS3 audiovisual dataset. To understand generalization capabilities of these models we evaluate them on an out-of-domain Audio-only dataset and also compare them with other state-the-art methods on this dataset. We use the well-established DNS Challenge eval set (Reddy et al., 2020) for this evaluation. There are a few points worth noting from Table 5. Unlike other methods in the table, our Audio-only is *not* trained on DNS Challenge set, yet the performance is competitive with other state-of-the-art methods. Moreover, our model is causal and relatively small, unlike some of the other models in the table. Overall, it shows that our Audio-only speech enhancement experimental backbone is a strong and competitive backbone model. The MUTUD model improves over the Audio-only model on this evaluation set as well. Furthermore, this independent evaluation on an Audio-only dataset shows that MUTUD generalizes well and is not limited to audiovisual data seen during training. It also evidences practicality and usefulness of MUTUD in real-word, as this test setting is how the model will be used in real-world.

## 5.3 Efficiency Analysis

Table 6 shows parameter and Multiply Accumulate Operations (MAC) counts for all models. The MUTUD model is comparable in size and compute to both Audio-only models. In fact, the MAC for MUTUD is around 13% lower compared to even the Audio-only model with a matching parameter count. However, we saw in Table 1 that MUTUD is much more superior compared to these models. With respect to the Audiovisual model, MUTUD is smaller almost by a factor of 5 and has a smaller size and MAC by around 83% and 77% respectively. This shows the massive gain in efficiency one can achieve through our MUTUD learning framework.

Table 5: Performance comparison on DNS Challenge evaluation (no-reverb) set.

| Method | PESQ | SI-SDR | STOI | # Params |
|---|---|---|---|---|
| **Noisy** | 1.58 | 9.07 | 0.92 | - |
| **FullSubnet** (Hao et al., 2021) | 2.77 | 17.29 | 0.96 | 5.6M |
| **CleanUNet** (Kong et al., 2022) | 3.15 | - | 0.96 | 46.07M |
| **Demucs** (Defossez et al., 2020) | 2.66 | - | 0.97 | 33.53M |
| **NSNet** (Xia et al., 2020) | 2.15 | 15.61 | 0.94 | 5.1M |
| **Audio-only** (Ours) | 2.32 | 16.24 | 0.94 | 2.98M |
| **MUTUD** (Ours) | 2.56 | 17.50 | 0.96 | 3.63M |

Table 6: Number of Parameters and Multiply Accumulate Operations (MACs) for all models.

| | **Audio-only** | **Audio-only** *w. matched params* | **Audiovisual** | **MUTUD** |
|---|---|---|---|---|
| **# of Param.** | 2.978M | 3.627M | 15.736M | 3.635M |
| **MACs** | 1.381G | 1.821G | 9.324G | 1.593G |
| **Inference Time (ms)** | $98.1 \pm 2.87$ | $100.2 \pm 2.50$ | $206.0 \pm 8.1$ | $108.0 \pm 2.1$ |

### 5.4 TAME Module Analysis

To analyze the estimated video features from the TAME module, we measure how similar they are to the original video and audio features. We compute the average cosine similarity and $\ell_2$ distance between video features and estimated video features ($F_v$ vs. $\hat{F}_v$), video features and audio features ($F_v$ vs. $F_a$), and estimated video features and audio features ($\hat{F}_v$ vs. $F_a$) for SNRs ranging from 5dB to -15dB. Figure 4 clearly indicates that the cosine similarity between the estimated video features and the original video features is high, around 0.94, while the similarity between audio and original (estimated) video features is low, $\approx -0.40$ ($\approx -0.42$). The $\ell_2$ distances show a similar trend where the audio and video features are further apart, and the estimated video features and the original ones are much closer. The high similarity between the estimated and original video features, while having low similarity between the estimated video and audio features evidence that TAME is not just regurgitating audio features but is actually functioning as designed (use audio information to get video information).

In addition, in Figure 5 we show the t-SNE visualization of the estimated video features, the original video features, and the audio features for all SNRs. Analyzing the clusters, we can clearly observe that the audio features $F_a$ form a distinct group, separate from the estimated video features $\hat{F}_v$ and the actual video feature $F_v$, demonstrating that the TAME can differentiate between modality-specific characteristics. More importantly, the estimated video features $\hat{F}_v$ and the actual video features $F_v$ are clustered closely together in the feature space, implying that the TAME module can accurately retrieve video features from the memory block, closely mirroring the actual video features even as the SNR levels decrease.

We further analyze the distribution across the audio and video codebooks for all $K$. Figure 6 shows a visualization of all the learned codebooks $\boldsymbol{C}^a$ and $\boldsymbol{C}^v$ for audio and video. We visualize the mean of all 32 codebooks for each $k$. One key inference from this visualization is that all codes are well-represented and the training formulation does not lead to mode collapse. This is further evidenced through the visualization of the probability distribution $q$ for a sample noisy audio frame in Figure 7. Figure 7 shows the variation in the usage pattern of different codebooks by just a single frame of audio and shows that the codes are not collapsing and are learning to capture the expected information.

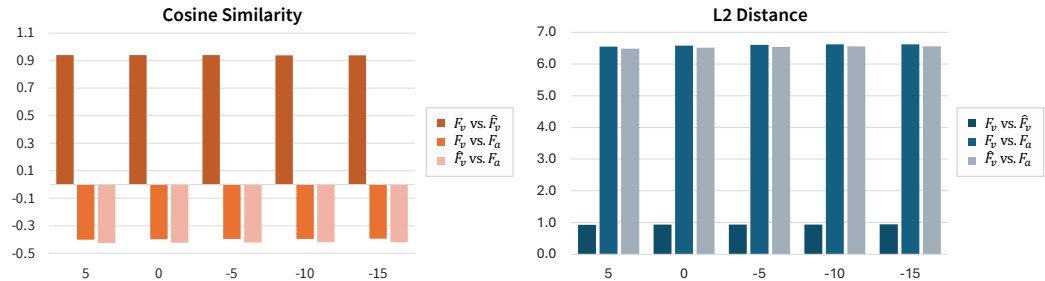

Figure 4: Cosine similarity (red) and $\ell_2$ distance (blue) between video features and estimated video features, video and audio features, and estimated video and audio features for different SNRs.

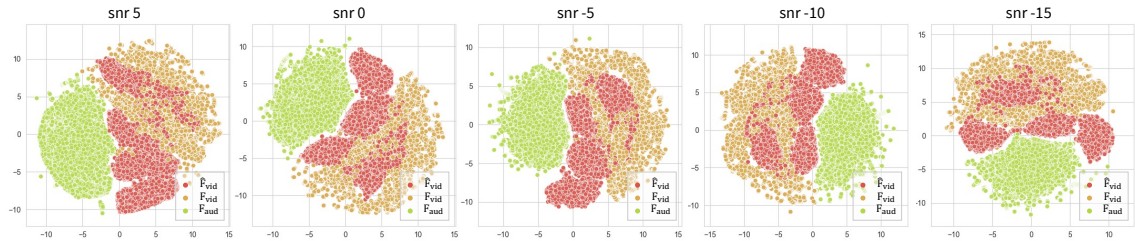

Figure 5: TSNE visualization of the estimated video features $\hat{F}_v$, the actual video features $F_v$, and the audio features $F_a$ for SNRs ranging from 5dB to -15dB.

## 5.5 Ablation for Codebook Size

We perform an ablation for the size of the codebooks in MSCs. We experiment with 4 different codebook sizes, $N$ (8, 16, 32, and 64) in each MSC of the TAME module. Table 7 indicates that as the size increases, more gain in speech enhancement performance is achieved, meaning that a larger number of codes in the MSCs can contain more meaningful features. We see that the $N = 64$ does not get much performance gain over $N = 32$. $N = 32$ is sufficient for embedding the audio and video information into the codebooks and then relating them to enable estimation of video representations using audio.

Table 7: Ablation on different numbers of codes, $N$, in each MSC of TAME.

| # of codes | STOI (%) | | | | | SISDR (dB) | | | | | PESQ | | | | |
|---|---|---|---|---|---|---|---|---|---|---|---|---|---|---|---|
| | 5 | 0 | -5 | -10 | -15 | 5 | 0 | -5 | -10 | -15 | 5 | 0 | -5 | -10 | -15 |
| 8 | 93.3 | 88.7 | 80.9 | 68.6 | 53.0 | 13.91 | 10.71 | 7.20 | 3.02 | -2.39 | 2.36 | 1.91 | 1.56 | 1.32 | 1.17 |
| 16 | 93.4 | 88.8 | 81.1 | 68.9 | 53.3 | 13.98 | 10.81 | 7.30 | 3.10 | -2.41 | 2.35 | 1.92 | 1.57 | 1.32 | 1.17 |
| 32 | 93.5 | **89.2** | **81.8** | **69.8** | **54.0** | **14.11** | **11.02** | **7.56** | **3.38** | **-2.19** | 2.36 | 1.92 | 1.57 | 1.32 | 1.17 |
| 64 | **93.6** | 89.1 | 81.6 | 69.6 | **54.0** | **14.11** | 11.00 | 7.51 | 3.31 | -2.21 | **2.38** | **1.95** | **1.59** | **1.34** | **1.18** |

Table 8: MUTUD effectiveness in Audiovisual Speech Recognition (AVSR) task.

| Method | WER (%) ↓ | | | | |
|---|---|---|---|---|---|
| | 5 | 0 | -5 | -10 | -15 |
| Audio-only | 12.24 | 17.836 | 31.37 | 60.64 | 93.32 |
| Audiovisual | **5.26** | **7.088** | **11.01** | **21.12** | **36.56** |
| MUTUD | 11.71 | 16.299 | 24.99 | 44.07 | 73.56 |

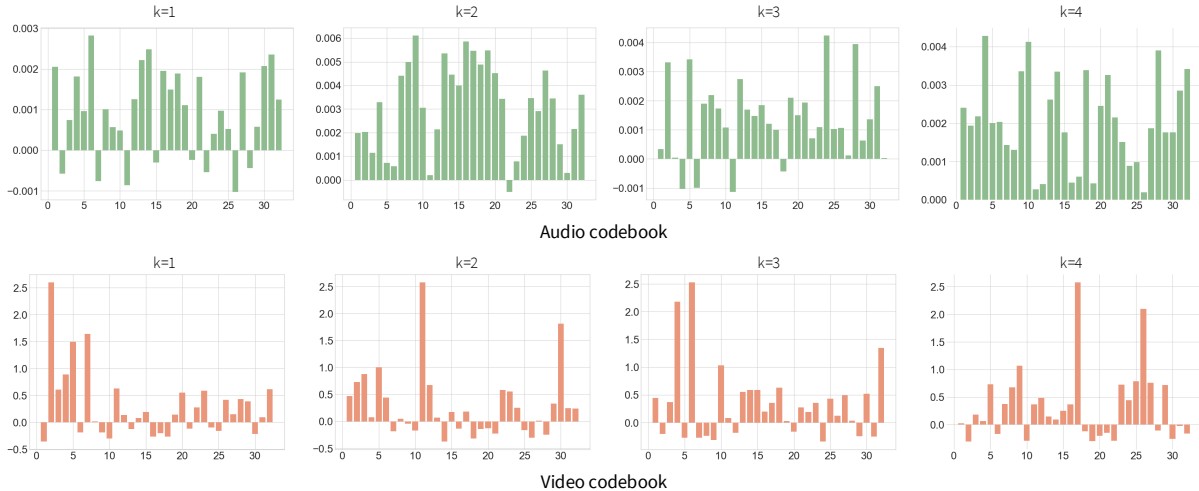

Figure 6: Visualization of the learned audio and video codebooks ($\boldsymbol{C}^a$ and $\boldsymbol{C}^v$). The plots show the mean of each code in all $K(=4)$ codebooks.

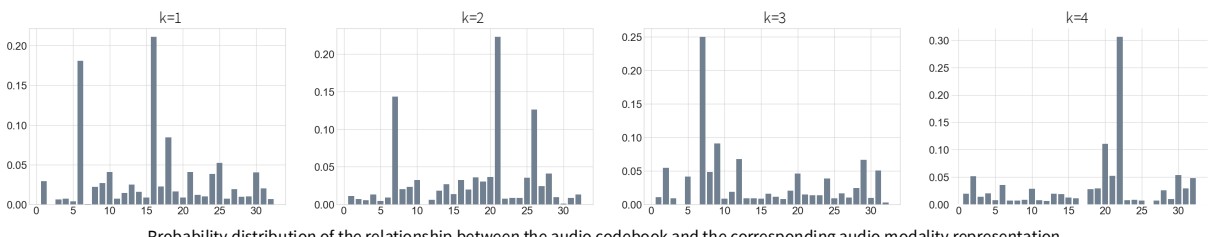

Figure 7: Visualization of the probability distribution $q$ (for each $k$) for a sample noisy audio frame.

## 5.6 Audiovisual Speech Recognition (AVSR)

To showcase our method's versatility, we additionally show results on different downstream tasks: AVSR and AV-ASD to verify the effectiveness of the proposed TAME module. In this subsection, we will show the experimental details for the AVSR task and the performance analysis.

### 5.6.1 Implementation Details

For AVSR we adopt V-CAFE (Hong et al., 2022) as a baseline architecture. The video encoder in the V-CAFE architecture consists of a 3D convolution layer with Batch Normalization and Max-pool followed by ResNet-18 (He et al., 2016), and the audio encoder contains two 2D convolution layers followed by one ResBlock for the audio front-end. The input shape of the model is the same as the Audiovisual Speech Enhancement model. VCAFE consists of a cross-modal attention followed by a noise reduction mask. The noise reduction mask consists of two convolution layers with ReLU and Sigmoid activation respectively. The mask is multiplicated to the audio features $f_a$, and the masked audio features are summed with the original features to obtain the enhanced audio features. Finally, with the enhanced audio features and the visual features are concatenated with the linear layer and taken into Conformer (Gulati et al., 2020) for the encoder and Transformer (Serdyuk et al., 2022) for the decoder for predicting the speech.

The Conformer (Gulati et al., 2020) sequence encoder is composed of hidden dimensions of 512, feed-forward dimensions of 2048, 12 layers, 8 attention heads, and a convolution kernel size of 31. The Transformer (Serdyuk et al., 2022) sequence decoder contains hidden dimensions of 512, feed-forward dimensions of 2048, 6 layers, and 8 attention heads are employed. Note that the video features are upsampled with the nearest neighbor interpolation to match the size of the audio features when taken into the proposed TAME Module.

Table 9: MUTUD effectiveness in Audiovisual Active Speaker Detection (AV-ASD) task.

| Method | mAP(%) |
|---|---|
| SyncNet (Chung & Zisserman, 2017) | 82.1 |
| TalkNet (Tao et al., 2021) | 79.9 |
| SPELL+ (Min et al., 2022) | 85.9 |
| **Audiovisual EgoASD** (Huh et al., 2025) | **87.6** |
| **Video-only EgoASD** | **82.3** |
| **MUTUD EgoASD** | **86.5** |

V-CAFE achieves a Word Error Rates (WER) of 2.9% on LRS3 tests when trained on only LRS3 dataset, which is competitive with other works such as (Ma et al., 2021b). Other works such as (Shi et al., 2022; Serdyuk et al., 2022) use much larger additional training data for slightly better WER. This shows that V-CAFE is a simple and reliable backbone for uses in our experiments.

To make results more insightful we show results under noisy conditions. We utilize background noises in diverse environments of DEMAND (Thiemann et al., 2013) dataset with SNR range randomly chosen from -15dB to 15dB for training. For testing, we report the testing performance at five different SNRs (in dB): 5, 0, -5, -10, and -15, measuring speech recognition quality through (WER).

### 5.6.2 Performance Analysis

As shown in Table 8, MUTUD, while not outperforming the AV approach, shows a substantial reduction in WER compared to the Audio-only method, highlighting TAME module's contribution in learning to leverage visuals even if it is available only during training. This is especially true for low-SNRs where visuals play more important roles and MUTUD can help reduce WER by a considerable margin (6% for -5dB and 16% for -10dB in absolute terms). The reduced WER across various levels of background noise indicates that the TAME module effectively utilizes visual information to complement audio input, thus enhancing overall speech recognition accuracy.

### 5.7 Audiovisual Active Speaker Detection (AV-ASD)

We additionally conduct the experiment on the AV-ASD task. In the AV-ASD task, we assume the absence of audio modality at inference time, instead of the visual modality as done in previous experiments. We show results on the EasyCom dataset, a considerably more challenging real-world noisy dataset than the LRS3-TED dataset.

### 5.7.1 Implementation Details

The AV-ASD model follows Huh et al. (2025). It consists of a mouth keypoint detector to crop the lip region, CNN-based video, and audio encoders, and a fusion layer followed by a causal temporal layer to incorporate a longer temporal past context. For the mouth keypoint detector, we adopt the ground truth facial per speaker manually checked by annotators. From the keypoints, we generate a new face crop by cropping the region by half of the width of the face region horizontally and also crop a quarter of the height downwards and three-quarters of the height upwards from the center of the mouth. We also generate a lip region, cropping the same way horizontally but cropping a quarter of height up and down from the mouth center.

The audio encoder is adopted from a VGG-M (Chatfield et al., 2014) operating on 13-dim Mel-Frequency Cepstral Coefficient (MFCC). For the video encoder, we use a spatio-temporal VGG-M (Chatfield et al., 2014) composed of a 3D convolutional layer followed by a stack of 2D convolutions. We also adopt a Self-Attentive Pooling (SAP) layer (Bhattacharya et al., 2017) for fusing the output audio and video features. Lastly, we set a unidirectional LSTM layer for temporal modeling to sequentially process consecutive embeddings from the fusion layers to predict speech activity corresponding to the latest frame followed by a projection layer and a sigmoid activation to derive activity predictions for each target speaker. For TAME module integration, like

the AVSR model, the video features are upsampled with the nearest-neighbor interpolation to match the number of the audio features.

For training, we apply horizontal flipping, random rotation within $-15° \sim +15°$, and motion blur augmentation with kernels randomly from 10, 25, 50, and 100. Due to the limited size of Easycom, we firstly pretrain the model with a larger dataset, VoxCeleb2 (Chung et al., 2018), to produce a better performance and generalization. We train using SGD optimizer (Robbins & Monro, 1951) with a learning rate of $5^{-5}$ with a weight decay of $5^{-4}$. We evaluate the performance using the mean Average Precision (mAP).

### 5.7.2 Performance Analysis

As indicated in Table 9, the model showcases a mean Average Precision (mAP) of 86.50%, which sits comfortably between the Video-only method at 82.25% and the AudioVisual approach at 87.60%. Notably, this outcome demonstrates the TAME's capability to properly retrieve audio features, complementing the previously illustrated proficiency in video feature retrieval. Therefore, the comparative performance indicates that the proposed TAME is not only effective in leveraging visual information but also exhibits a reciprocal competence in audio feature retrieval, thereby reinforcing its applicability in multimodal scenarios. Thus, the experimental results underscore the versatility of the proposed TAME module.

## 6 Conclusion

This work is motivated to address practical challenges in using multimodal solutions in real-world applications. We build and train the models keeping in mind that inference will be unimodal – a multimodal training but unimodal deployment strategy, and propose MUTUD. In MUTUD, the model learns to associate and relate different modalities through modality-specific codebooks. Once this is achieved during training, the representations of modality absent during inference are obtained using the one present during inference. We show substantial gains over corresponding unimodal models and efficiency gains over full multimodal counterparts while retaining performance to a considerable extent. Moreover, our framework and TAME are fairly generic and can be easily adapted for other common multimodal learning tasks and models. We can also extend MUTUD to more than two modalities through pairwise MSCs.

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
