# OpenReview forum: "Efficient Audiovisual Speech Processing via MUTUD: Multimodal Training and Unimodal Deployment"
_TMLR — Accepted by TMLR_

### Review · Reviewer_Hswm · 2025-09-28

**Summary Of Contributions:**

This paper tackles the significant discrepancy between the multimodal training and unimodal deployment of speech processing systems. The authors propose MUTUD, a framework designed to retain the benefits of multimodal learning in a practical, unimodal inference setting. The core of this framework is the Temporally Aligned Modality feature Estimation (TAME) module, which employs modality-specific codebooks to learn cross-modal associations during training. At deployment, TAME can then estimate the features of a missing modality (e.g., video) using only the available modality (e.g., audio), thus bridging the performance gap while maintaining unimodal efficiency.
1) Strengths
- The paper studies a critical and practical problem: the discrepancy between training and deployment in audiovisual speech processing.
- The effectiveness of the proposed MUTUD is well-supported by extensive experiments and in-depth analyses across multiple audiovisual tasks and datasets.

2) Weaknesses
- The core idea, using a learnable module to impute missing modalities, is not new. For example, DCP[1] directly adopts a neural network to predict missing modalities via contrastive learning.
- The proposed framework introduces too many hyperparameters (loss weights $\alpha, \beta, \gamma, \lambda$, codebook size $K$, and temperature $\tau$). This complexity, coupled with the lack of tuning guidance, may pose a challenge for real-world applications.

[1] Lin, Yijie, et al. "Dual contrastive prediction for incomplete multi-view representation learning." IEEE Transactions on Pattern Analysis and Machine Intelligence 45.4 (2022): 4447-4461.

**Additional Comments:**

N/A

**Audience:**

Yes

**Audience Explanation:**

The paper addresses the "gap between training and deployment", a fundamental and practical problem in machine learning where models that are powerful during training are infeasible for deployment. The proposed MUTUD framework is presented as a generic approach that could be adapted to other multimodal domains beyond speech processing (e.g., video-text, audio-text), making the idea highly valuable to the wider community.

**Claims And Evidence:**

Yes

**Claims Explanation:**

Performance Improvement: Tables 1, 2, 3, 4 (Speech Enhancement), Table 8 (Speech Recognition), and Table 9 (Active Speaker Detection) consistently show that MUTUD outperforms the audio-only baselines by a significant margin across all evaluation metrics.

 Inference Efficiency: The claim of high efficiency at inference is clearly demonstrated in Table 6. MUTUD has a parameter count and computational cost that are comparable to the unimodal model, while being drastically more efficient than the full audiovisual model.

**Requested Changes:**

The paper would be strengthened by an ablation study on the components of the TAME loss function. The contribution of each term, particularly the cross-modal association loss $\mathcal{L}_{C_a\rightarrow C_v}$, is not experimentally justified. Such an analysis would provide better insight into the design of the training objective.

Key hyperparameters, such as the loss weights ($\alpha, \beta, \gamma, \lambda$) and the softmax temperature ($\tau$), are simply set to fixed values without justification. The absence of a sensitivity analysis makes it unclear how robust the model is to these choices and offers limited guidance for researchers aiming to reproduce or adapt the method for other applications.

---

> ### Author Response · Authors · 2025-10-25
> **Response to Reviewer Hswm**
>
> Thank you for your thoughtful feedback.
>
> **Requested Changes**
>
> - **Cross-modal association loss**
>
>   We agree that understanding the contribution of each loss term is important. Conceptually, $\mathcal{L}\_{C_a \rightarrow C_v}$ ties the temporal code-selection distributions between audio and video, so that the audio-only pathway can reproduce video information at the correct time indices during inference. This term is essential for producing video-like features consumed by the decoder and is a key objective in TAME. Given its role in enforcing temporal distributional alignment, removing $\mathcal{L}\_{C_a \rightarrow C_v}$ would collapse the estimation pathway toward frequent or average codes, thus incompatible with unimodal deployment.
>
> - **Sensitivity to hyperparameters**
>
>   $\alpha$, $\beta$, and $\gamma$ do not require tuning; for our experiments, all are set to one. To improve clarity and simplicity, we have removed them in the revised manuscript. $\lambda$ is task-specific, may be the only parameter that may need to be scaled depending on the task.

---

### Review · Reviewer_fbgS · 2025-10-15

**Summary Of Contributions:**

This paper introduce a multi-modal alignment between the video and audio so that the model can leverage one or a few modalities during the inference. Extensive experiments are conducted to validate the performance of the algorithm

Strength:
- The idea of leveraging the video to assist the audio speech processing is interesting and the authors demonstrated the performance improvement
- The organization and formulation of the problem setup is clear.

Weakness:
- Multimodal alignment with a CLIP-like loss is not new and it's not suprising to see that the modalities after the alignments can be used without the help of the other modalities.
- The elaboration on the temporal information is not clear and needs further clarification.

**Audience:**

Yes

**Audience Explanation:**

This is aligned with the TMLR audience's interest.

**Claims And Evidence:**

Yes

**Claims Explanation:**

The claim is accurate with clear formulation.

The only concern is that the math formulations are kind of messy to read, for example, the subscript on the left side always confusing my reading.

**Requested Changes:**

- math formulations are kind of messy to read, for example, the subscript on the left side always confusing my reading, also it would be helpful changing the inner product from $<,>$ to $\langle, \rangle$
- The temporal information should be highlighted besides the regular modality alignment
- Experiments should be highlighted with the best response in boldface to help understanding the performance difference.
- Needs to highlight the difference between the modality alignment and the representation learning based on these related works.

---

> ### Author Response · Authors · 2025-10-25
> **Response to Reviewer fbgS**
>
> Thank you for your positive evaluation and thoughtful suggestions.
>
>
> **Requested Changes**
>
> - **Math formulations**
>
>   We have updated the math formulations in the revised paper to make it more readable.
> - **Highlighting temporal alignment in TAME**
>
>   We have made changes in the text to highlight temporal alignment along with modality alignment (text in blue in Sec 1). This goes well with the mathematical description of TAME provided in Sec. 3.2.
> - **Best response in boldface**
>
>   Updated in the revised paper.
> - **Modality alignment and representation learning**
>
>   Our approach differs from generic alignment (e.g., CLIP-style objectives) and from broad representation learning. Alignment brings heterogeneous modalities close in a shared space but offers no inference-time route to reconstruct a missing modality; representation learning seeks general features, not a deployment-time replacement for an absent encoder. In contrast, MUTUD/TAME uses temporally coupled, modality-specific codebooks with distribution matching to estimate the missing modality and feed it to the decoder, enabling multimodal training with unimodal deployment while preserving most AV gains.

---

### Review · Reviewer_2T6j · 2025-10-16

**Summary Of Contributions:**

The paper introduces a new framework called Multimodal Training and Unimodal Deployment (MUTUD), which allows models to leverage information from multiple modalities, such as audio and visual data, during training, but requires only a single modality during inference. Central to this approach is the Temporally Aligned Modality feature Estimation (TAME) module, which can estimate the features of missing modalities using information from the available ones at inference time. By using cross-modal codebooks, the TAME module enables the model to recall and reconstruct representations of absent inputs. This framework addresses key challenges in deploying multimodal systems, such as increased computational demands, complex data synchronization, and the need for multiple sensors during inference, by allowing unimodal deployment without significant loss in performance. The paper demonstrates that MUTUD can substantially narrow the performance gap between unimodal and multimodal models, while also achieving considerable reductions in model size and computational cost—sometimes by as much as 80% compared to full multimodal models. The approach is validated across several speech processing tasks, including speech enhancement, speech recognition, and active speaker detection, and consistently shows improved unimodal inference compared to models trained only on single modalities.

**Additional Comments:**

N/A

**Audience:**

Yes

**Audience Explanation:**

This method can also be extended to a wider range of audio-visual applications, such as video understanding, where one modality may be unavailable during real-world deployment.

**Claims And Evidence:**

Yes

**Claims Explanation:**

1. The paper introduces codebooks that effectively link features between visual and audio modalities, enabling the estimation of missing features during inference to compensate for the absence of one modality.

2. Experimental results demonstrate that the proposed model can reduce computational costs by up to 80% in certain cases.

3. The experiments, as presented in Tables 1, 2, and 3, show that the method significantly improves performance even when only a single modality is available during inference.

**Requested Changes:**

The paper lacks relevant visualizations to illustrate the improvements over the baseline. The authors could enhance their analysis by visualizing the spectrograms of both the predictions and the ground-truth, providing a clearer comparison of model performance.

---

> ### Author Response · Authors · 2025-10-25
> **Response to Reviewer 2T6j**
>
> Thank you for your helpful feedback.
>
> **Spectrogram visualization**: As per the requested change, we have added spectrogram visualization to compare different methods across different SNRs in the revised paper (Sec. 5.1, Figure 3). The advantage of MUTUD over Audio-only is most visible at SNR −10/−15 dB: MUTUD preserves thinner, more continuous harmonics and darker inter-harmonic valleys (lower noise floor), and it shows a cleaner F0 trajectory with reduced low-frequency clutter. At −5 dB, MUTUD further reduces high-frequency speckle and keeps consonant transitions (short vertical stripes) sharper than Audio-only. Across all SNRs, MUTUD shows results consistently more similar to Audiovisual than Audio-only.

---

### Decision · Action_Editor_jBjT · 2025-12-04

**Recommendation:** Accept with minor revision

**Audience:**

Yes

**Audience Explanation:**

This multimodal training and inference framework has good potential for the multimodal community.

**Claims And Evidence:**

Yes

**Claims Explanation:**

1) A clear and practical framework for exploiting multimodal information during training while enabling efficient unimodal deployment.
2) The proposed TAME module effectively reconstructs missing-modality features through cross-modal codebooks, substantially reducing computation and sensor requirements without sacrificing accuracy. The approach is well-motivated, addresses a real deployment bottleneck, and generalizes across multiple speech tasks, consistently narrowing the unimodal–multimodal performance gap.
3) The problem setup is clearly formulated, and experiments demonstrate strong gains, showing the method’s practical value and potential applicability to broader multimodal domains.

The only issue is the hyperparameter issue raised by reviewer 3. Even though I understand that in today's models, hyperparameters are everywhere, I hope that the authors may open-source their model for reproducibility and further analyze the sensitivity of the hyperparameter.